# The dynamics and longevity of circulating CD4+ memory T cells depend on cell age and not the chronological age of the host

**M. Elise Bullock**[1⊙], **Thea Hogan**[2⊙], **Cayman Williams**[2⊙], **Sinead Morris**[1], **Maria Nowicka**[1], **Minahil Sharjeel**[2], **Christiaan van Dorp**[1], **Andrew J. Yates**[1‡]*, **Benedict Seddon**[2‡]*

**1** Department of Pathology and Cell Biology, Columbia University Irving Medical Center, New York, New York, United States of America, **2** Institute of Immunity and Transplantation, Division of Infection and Immunity, UCL, Royal Free Hospital, London, United Kingdom

⊙ These authors contributed equally to this work.
‡ AJY and BS also contributed equally to this work.
* andrew.yates@columbia.edu (AJY); benedict.seddon@ucl.ac.uk (BS)

**Data Availability Statement:** The mathematical models and our approach to model fitting are detailed in Text A and Text B of S1 File,

## Abstract

Quantifying the kinetics with which memory T cell populations are generated and maintained is essential for identifying the determinants of the duration of immunity. The quality and persistence of circulating CD4 effector memory ($T_{EM}$) and central memory ($T_{CM}$) T cells in mice appear to shift with age, but it is unclear whether these changes are driven by the aging host environment, by cell age effects, or both. Here, we address these issues by combining DNA labelling methods, established fate-mapping systems, a novel reporter mouse strain, and mathematical models. Together, these allow us to quantify the dynamics of both young and established circulating memory CD4 T cell subsets, within both young and old mice. We show that that these cells and their descendents become more persistent the longer they reside within the $T_{CM}$ and $T_{EM}$ pools. This behaviour may limit memory CD4 T cell diversity by skewing TCR repertoires towards clones generated early in life, but may also compensate for functional defects in new memory cells generated in old age.

## Introduction

CD4 T cells can suppress pathogen growth directly and coordinate the activity of other immune cells [1,2]. Following exposure to antigen, heterogeneous populations of circulating memory CD4 T cells are established that enhance protection to subsequent exposures. In mice, these can be categorised broadly as central ($T_{CM}$) or effector ($T_{EM}$) memory. Canonically defined $T_{CM}$ have high expression of CD62L, which allow them to circulate between blood and secondary lymphoid organs, while CD4 $T_{EM}$ express low levels of CD62L and migrate to tissues and sites of inflammation. The lineage relationships between CD4 $T_{CM}$, $T_{EM}$, and other memory subsets remain unclear [3].

In mice, circulating memory CD4 T cells appear not to be intrinsically long-lived on average but are lost through death or onward differentiation on timescales of days to weeks [4–9].

respectively. All code and data used to perform model fitting, and details of the prior distributions for parameters, are available at https://github.com/elisebullock/PLOS2024 and also at doi:10.5281/zenodo.11476381.

**Funding:** This study was funded by the National Institutes of Health (R01 AI093870 and U01 AI150680 to AJY) and the Medical Research Council (MR/P011225/1 to BS). The funders played no role in the study design, data collection and analysis, decision to publish, or preparation of the manuscript.

**Competing interests:** The authors have declared that no competing interests exist.

**Abbreviations:** BMT, bone marrow transplant; BrdU, bromodeoxyuridine; ELPD, expected log pointwise predictive density; HSC, haematopoietic stem cell; LOO, Leave-One-Out; MP, memory phenotype; RTE, recent thymic emigrant; TCR, T cell receptor; WT, wild-type; YFV, yellow fever virus.

However, compensatory self-renewal can act to sustain memory T cell clones, defined as populations sharing a given T cell receptor (TCR), for much longer than the life span of their constituent cells [10]. To understand in detail how the size of memory T cell populations change with time since exposure to a pathogen, we therefore need to be able to quantify their rates of de novo production, division, and loss; any heterogeneity in the rates of these processes; how kinetically distinct subsets are related; and how these kinetics might change with host and/or cell age.

Much of our knowledge of memory CD4 T cell dynamics derives from studies of laboratory mice housed in specific pathogen-free conditions, which have not experienced overt infections, but nevertheless have abundant memory phenotype (MP) CD4 T cells [11–13]. These populations contain both rapidly dividing and more quiescent populations [4,14–17], a stratification that holds within the $T_{CM}$ and $T_{EM}$ subsets [5]. MP T cells are established soon after birth and attain broadly stable numbers determined in part by the cleanliness of their housing conditions [18]. After the first few weeks of life, however, both CD4 $T_{CM}$ and CD4 $T_{EM}$ are continuously replaced at a rates of a few percent of the population size per day [5], independent of their housing conditions [18]. MP CD4 T cells are therefore likely specific for an array of commensals, self, or ubiquitous environmental antigens and may comprise subpopulations at different stages of maturation or developmental end points. Dissecting their dynamics in detail may then shed light on the life histories of conventional memory cells induced by infection.

A widely used approach to studying lymphocyte population dynamics is DNA labelling, in which an identifiable agent such as bromodeoxyuridine (BrdU) or deuterium is taken up during cell division. By measuring the frequency of cells within a population that contain the label during and following its administration, one can use mathematical models to extract rates of production and turnover (loss). Without allowing for the possibility variation in these rates within a population, their average estimates may be unreliable [19,20]. Explicit treatment of such "kinetic heterogeneity" can yield estimates of average production and loss rates that are interpretable and independent of the labelling period, although the number of distinct subsets and the lineage relationships between them are difficult to resolve [4,5,18,21,22]. Further, models of labelling dynamics usually conflate the processes of influx of new cells and self-renewal of existing ones into a single rate of cell production. Distinguishing these processes is particularly important in the context of memory, because their relative contributions determine the persistence of clones within a population; for a population at equilibrium, clones are diluted by influx but sustained by self-renewal.

To address these uncertainties, one can combine DNA labelling with information from other systems to validate predictions or constrain the choice of models. One approach is to obtain an independent estimate of any constitutive rate of flow into a population by following its replacement by labelled precursors. In a series of studies, we established a mouse model that yields the rates of replacement of all peripheral lymphocyte subsets at steady state [5,18,23–28]. In this system, illustrated in Fig 1A, low doses of the transplant conditioning drug busulfan are administered to specifically deplete haematopoietic stem cells (HSCs) in adult mice, while leaving peripheral lymphocyte compartments undisturbed. Shortly afterward, the treated mice receive T and B cell–depleted bone marrow from congenic donors, to reconstitute the HSC niche. Donor fractions of 70% to 90% are established rapidly among HSC, which are stable long term [23]. Within 6 weeks of transplant, the fraction of cells that are donor-derived (the chimerism) equilibrates at similar levels at all stages of thymic development and at the transitional stage of B cell development, indicating that turnover of all these populations is complete within this time frame [23]. Donor-derived cells proceed to gradually replace host-derived cells in all peripheral lymphocyte compartments over timescales of weeks to months, first

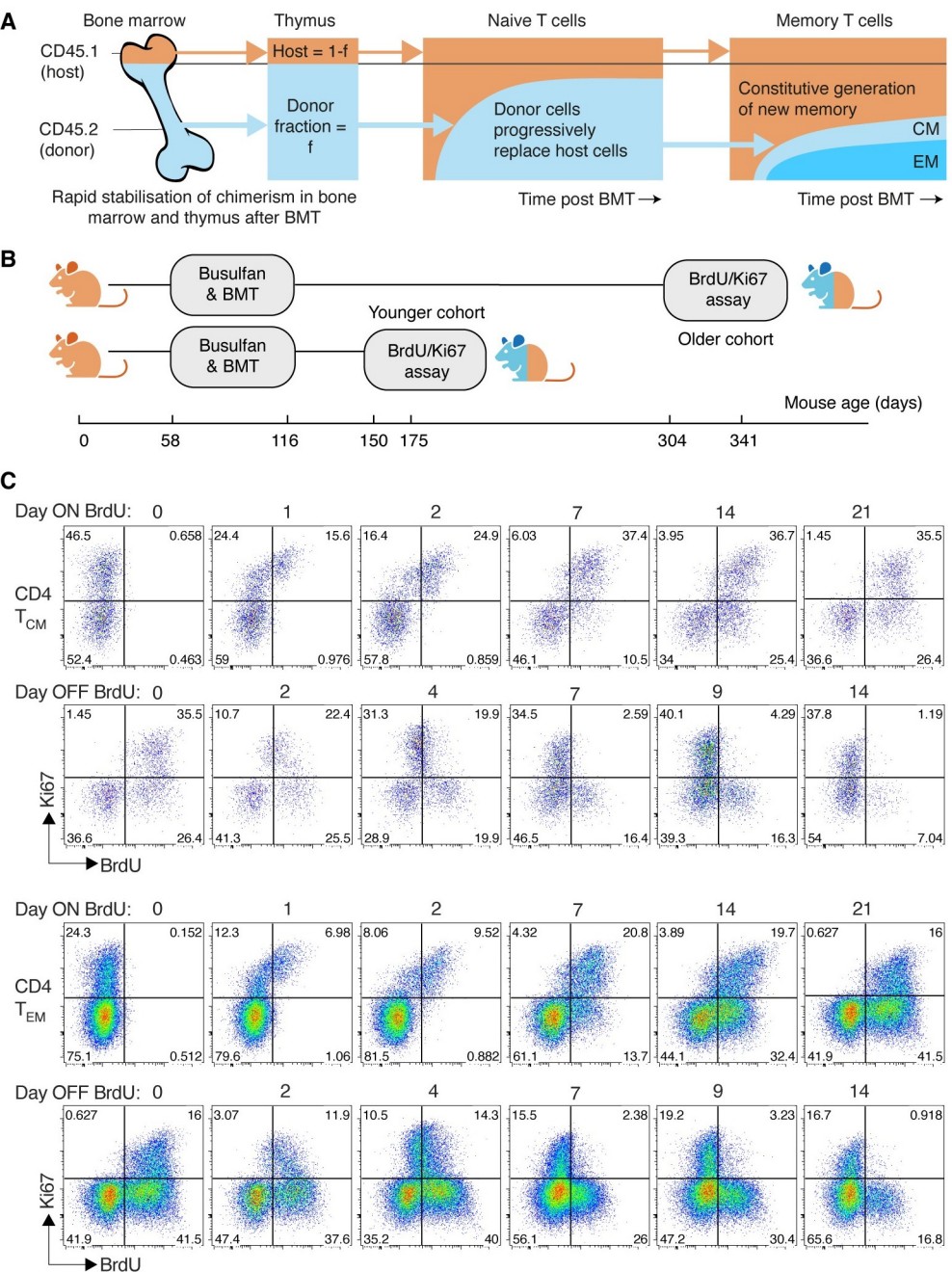

**Fig 1. Busulfan chimeric mice, experimental design, and gating strategies. (A)** Schematic of the busulfan chimeric mouse system, adapted from [32]. **(B)** Design of the BrdU/Ki67 labelling assay, studying 2 cohorts at different times post BMT. **(C)** Chimeric mice were fed BrdU for 21 days. Mice were analysed at different times during feeding and across 14 days following its cessation. Here, we show representative timecourses obtained from donor-derived CD4 $T_{CM}$ and $T_{EM}$, showing patterns of Ki67/BrdU expression during and after BrdU feeding. See Methods and Fig A of S1 File for details and gating strategies.

within naive populations and then more gradually within antigen-experienced subsets [5]. Importantly, total cell numbers within each subset remained indistinguishable from untreated controls at all timepoints [5], indicating that cells originating from the sex- and age-matched donor and host HSC behaved identically. By using mathematical models to describe the

timecourse of replacement within a particular population, one can obtain a direct estimate of the rate of influx of new cells, which can be fed directly into models of DNA labelling [5]. A conceptually similar approach to measuring influx is to use division-linked labelling to explicitly model the kinetics of label in both the population of interest and its precursor. This method is particularly useful when labelling is inefficient, because labelling among precursors is not saturated and is time-varying. Their influx then leaves an informative imprint on any labelling within the target population itself, allowing the rate of ingress of new cells to be inferred [29].

Another means of boosting the resolving power of DNA labelling assays is to use concurrent measurements of Ki67, a nuclear protein expressed during mitosis and for a few days afterwards [23,30,31]. A cell's BrdU content and Ki67 expression then report its historical and recent division activity, respectively. This approach was used to study MP CD4 T cell dynamics [5] and rule out a model of "temporal heterogeneity," in which CD4 $T_{CM}$ and $T_{EM}$ cells each comprise populations of cells that divide at a uniform rate, but that a cell's risk of loss varied with its expression of Ki67. The BrdU/Ki67 approach instead supported a model in which CD4 $T_{EM}$ and $T_{CM}$ each comprise at least 2 kinetically distinct ("fast" and "slow") subpopulations, consistent with other studies [4,16,17]. These subpopulations were assumed to form independent lineages that branch from a common precursor [5], but a subsequent study using the busulfan chimera system alone found support for an alternative model in which memory cells progress from fast to slow behaviour [18]. The topologies of the kinetic substructures of CD4 $T_{CM}$ and $T_{EM}$ therefore remain uncertain.

Here, we aimed to characterise the dynamics CD4 $T_{CM}$ and $T_{EM}$ in full, by combining these two approaches—performing BrdU/Ki67 labelling within busulfan chimeric mice directly. Our reasoning was three-fold. First, if one defines a memory cell's age as the time since it or its ancestor entered the memory pool, the average age of donor (newly recruited) clones will be lower than that of the host-derived clones that they are replacing. By stratifying the BrdU/Ki67 analyses by donor and host cells, one can therefore quantify any changes in the kinetics of memory T cell clones with their age. Second, by performing the labelling experiments in young and aged cohorts of mice, one can simultaneously observe any changes in dynamics due to the mouse's chronological age. Third, this combined approach could allow us to define the relationship between fast and slow memory CD4 T subsets and identify the immediate precursors of CD4 $T_{CM}$ and $T_{EM}$.

## Results

### Performing DNA labelling assays in busulfan chimeric mice

We studied 2 age classes of mice (Fig 1B). In one cohort, mice underwent busulfan treatment and bone marrow transplant (BMT) aged between 58 and 116 days, and BrdU/Ki67 labelling was performed 56 to 91 days later (at ages 150 to 175 days). For brevity, we refer to these as "young" mice. An "old" cohort underwent treatment and BMT at similar ages but with labelling performed between 246 and 260 days later (at ages 304 to 341 days). BrdU was administered by intraperitoneal injection and subsequently in drinking water for up to 21 days. Mice were killed at a range of time points within this period or up to 14 days after BrdU administration stopped. CD4 $T_{CM}$ and $T_{EM}$ were recovered from lymph nodes, enumerated, and stratified by their BrdU content and Ki67 expression (Fig 1C and Fig A of S1 File).

### The proliferative activity of CD4 $T_{CM}$ and $T_{EM}$ declines as they age

We first analysed the bulk properties of CD4 MP T cells within the two cohorts. The total numbers of CD4 $T_{CM}$ and $T_{EM}$ recovered from lymph nodes increased slightly with age

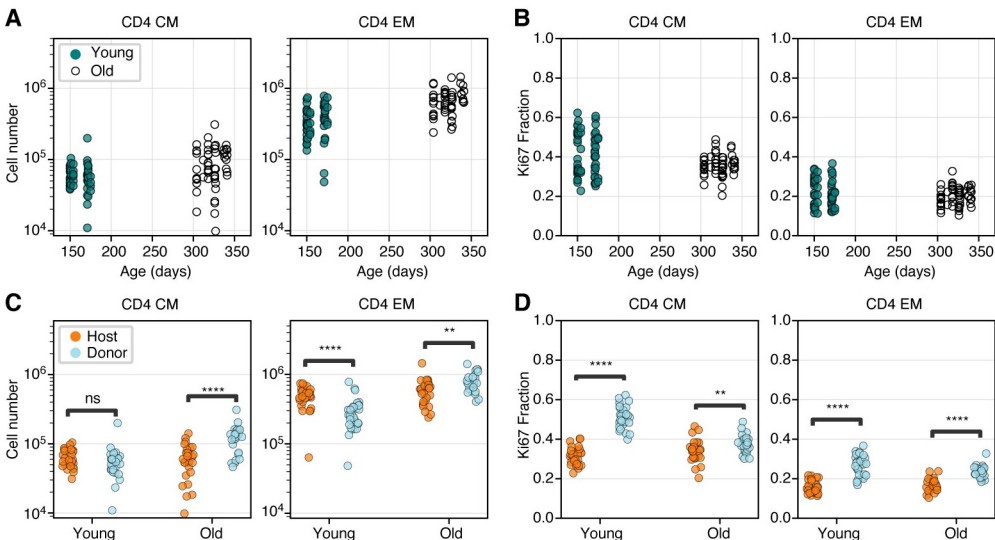

**Fig 2. Busulfan chimeric mice reveal shifts in memory CD4 T cell dynamics with both mouse age and cell age. (A, B)** Numbers and Ki67 expression levels of CD4 $T_{CM}$ and $T_{EM}$ recovered from lymph nodes in young and old cohort (Mann–Whitney tests). **(C, D)** Numbers and Ki67 expression levels of CD4 $T_{CM}$ and $T_{EM}$ in young and old mice, stratified by average cell age (host and donor; Wilcoxon tests). The data underlying this figure can be found in S1 Data. ** $p < 10^{-3}$; *** $p < 10^{-4}$; **** $p < 10^{-5}$.

(Fig 2A). Proliferative activity, measured by the proportion of cells expressing Ki67 (Fig 2B), fell slightly with age among CD4 $T_{CM}$ (median Ki67$^{high}$ fraction = 0.44 in young cohort, 0.36 in older cohort, $p = 0.02$) but not significantly for $T_{EM}$ (median Ki67$^{high}$ fraction = 0.20 in both age groups, $p = 0.48$).

Consistent with reports of the production of new MP CD4 T cells throughout life [5,18], and the progressive enrichment of all T cell subsets in donor-derived cells after BMT, the composition of both memory subsets shifted towards donor cells in older mice (Fig 2C). This shift was more marked for $T_{CM}$ ($p < 10^{-4}$) but was also significant for $T_{EM}$ ($p < 10^{-3}$). Strikingly, in the younger cohort, donor $T_{CM}$ and $T_{EM}$ both expressed Ki67 at substantially higher frequencies than their host counterparts (Fig 2D; $p < 10^{-8}$ both cases). These differences diminished with time post BMT but remained significant in the older cohort ($p < 10^{-3}$ for both $T_{CM}$ and $T_{EM}$).

These patterns of donor/host cell differences indicate that the average levels of proliferation within cohorts of CD4 $T_{CM}$ and $T_{EM}$ decline with the time since they or their ancestors entered the population. This effect is most apparent among $T_{CM}$ but still appreciable for $T_{EM}$.

## Modelling BrdU/Ki67 labelling kinetics in busulfan chimeric mice

To find a mechanistic explanation of these patterns, we then analysed the BrdU/Ki67 time-courses derived from the young and old mice. As described in Methods, and illustrated in Fig 1B, in each age cohort, we followed the uptake and loss of BrdU within CD4 $T_{CM}$ and $T_{EM}$, stratified by host and donor cells and by Ki67 expression, during 21-day pulse and 14-day chase periods. This strategy allowed us to isolate the dynamics of new and old memory cells within young and old mice.

We attempted to describe these data using three classes of mathematical model, shown schematically in Fig 3A. We employed two versions of a kinetic heterogeneity model: branched, in

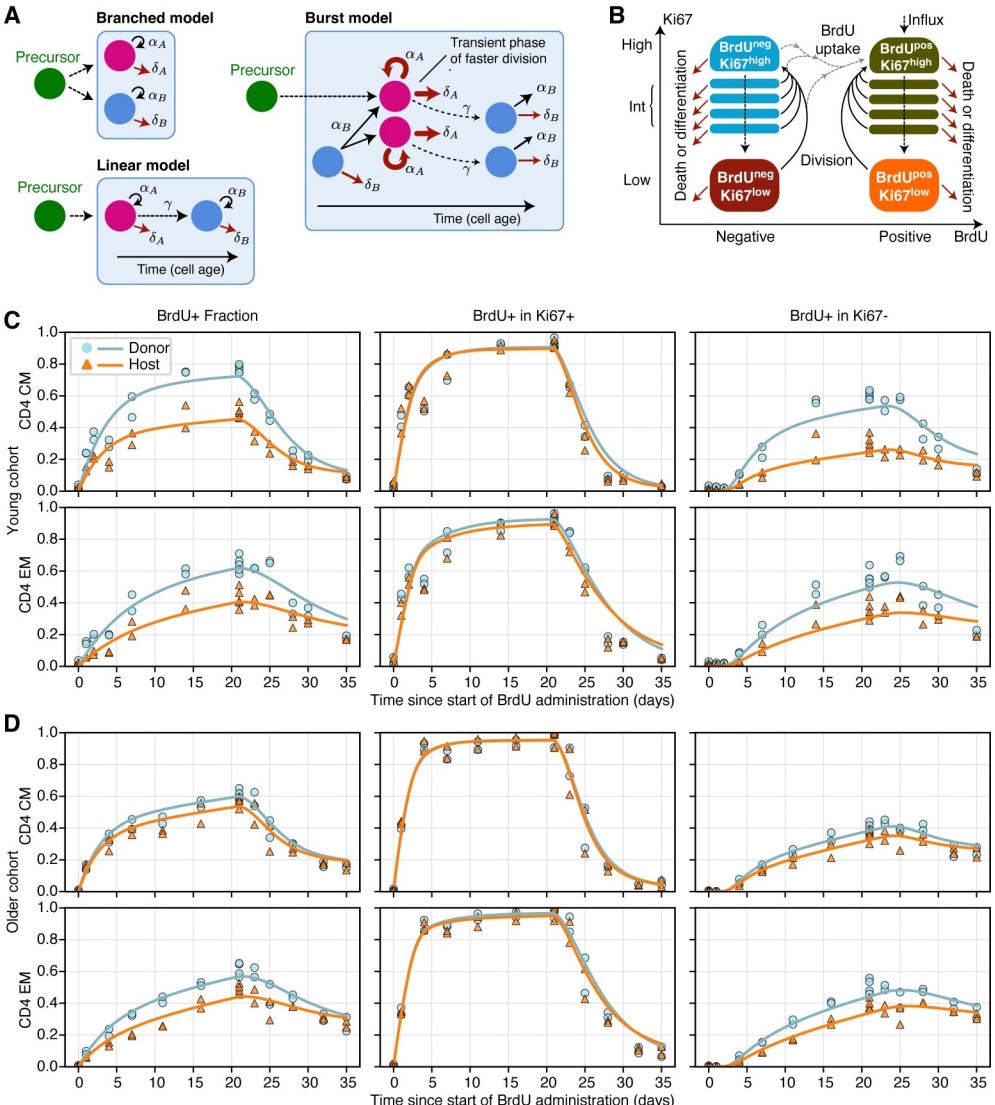

**Fig 3. Modelling BrdU/Ki67 timecourses. (A)** Models of heterogeneity in cellular dynamics within a memory subset. Adapted from [32]. **(B)** Schematic of the core component of the ODE model used to describe the uptake of BrdU and the dynamic of Ki67 expression. **(C, D)** Fitted kinetics of BrdU uptake and loss in CD4 $T_{CM}$ and $T_{EM}$, in the young **(C)** and older **(D)** cohorts of mice, using the branched model. The left column shows the frequency of BrdU-positive cells within donor and host cells. Centre and left columns show the proportion of BrdU-positive cells within Ki67-positive and Ki67-negative cells, respectively. There were 2 mice per time point for both donor and host throughout; in some panels, these points are very close and overlaid. The data underlying this figure can be found in S1 Data.

which newly generated $T_{CM}$ and $T_{EM}$ both immediately bifurcate into independent subpopulations (A and B) with distinct rates of division and death [5], and linear, in which cells enter memory with phenotype A and mature into a kinetically distinct phenotype B [18] (Fig 3A, models (ii) and (iii)). We also explored a "burst" model, describing a form of "temporal heterogeneity" [22], in which each memory subset ($T_{CM}/T_{EM}$, host/donor) comprises quiescent cells of type A, which are relatively long-lived and when triggered to divide enter a more dynamic state B, undergoing faster divisions with a reduced life span. These cells return to the quiescent state at some unknown rate. The burst model is a generalisation of the temporal heterogeneity

model we rejected previously [5], in which Ki67$^{high}$ cells, which are actively dividing or divided recently, are lost at a different rate to that of more quiescent Ki67$^{low}$ cells. The burst model is also conceptually similar to that explored in [33], in which a cell's risk of death changes with the time since its last division. In all the models, new cells entering memory are assumed to be Ki67$^{high}$, reflecting the assumption that they have recently undergone clonal expansion.

We encoded these models with systems of ordinary differential equations. The core structure, which describes the flows between BrdU$^{\pm}$ and Ki67$^{high/low}$ populations as they divide and die during the labelling period, is illustrated in Fig 3B. A detailed description of the formulation of the models is in Text A of S1 File.

Within both cohorts of mice, the total numbers of host and donor CD4 T$_{CM}$ and T$_{EM}$ cells, and the frequencies of Ki67$^{high}$ cells within each, were approximately constant over the course of the labelling assays (Fig B of S1 File). We therefore made the simplifying assumption that both donor and host memory populations were in quasi-equilibrium, with any changes in their numbers or dynamics occurring over timescales much longer than the 35-day labelling experiment. We took a Bayesian approach to estimating model parameters and performing model selection, using Python and Stan, which is detailed in Text B of S1 File. The code and data needed to reproduce our analyses and figures can be found at github.com/elisebullock/PLOS2024.

## Multiple models of memory CD4 T cell dynamics can explain BrdU/Ki67 labelling kinetics

The BrdU labelling data are summarised in Fig 3C and 3D, shown as the accumulation and loss of BrdU within each population as a whole (left hand panels) and within Ki67$^{high}$ and Ki67$^{low}$ cells (centre and right panels). In the young cohort (Fig 3C), the labelling kinetics of more established (host) and more recently generated (donor) CD4 T$_{CM}$ and T$_{EM}$ were quite distinct, with slower BrdU uptake within host-derived populations. In addition, heterogeneity was apparent in all labelling timecourses. For a kinetically heterogeneous population, the slope of this curve declines as the faster-dividing populations become saturated with label, and the majority of new uptake occurs within cells that divide more slowly. Such behaviour was apparent, consistent with the known variation in rates of division and/or loss within both T$_{CM}$ and T$_{EM}$. In the older cohort (Fig 3D), BrdU uptake was slower, differences between donor and host kinetics were less apparent, and the labelling kinetics of host-derived cells were similar to those in the younger cohort.

Intuitively, the initial rate of increase of the BrdU-labelled fraction in a population will depend both on the mean division rate of the population and the efficiency of BrdU incorporation per division [34]. This efficiency, which we denote $\epsilon$, is unknown, and this uncertainty can confound the estimation of other parameters [34,35]. In this regard, the timecourses of the proportion of Ki67$^{high}$ cells that are BrdU-positive (Fig 3C and 3D, centre panels) are highly informative. Any deviation from a rapid increase to 100% places bounds on $\epsilon$, and as we see below, the posterior distributions of this parameter were narrow. Similarly, the kinetics of the BrdU-positive fraction within Ki67$^{low}$ cells reflect the lifetime of Ki67 expression—we expect this lifetime to be the delay before the first BrdU-positive Ki67$^{low}$ cells appear, which is clearly apparent in the data (Fig 3C and 3D, right hand panels) and was again tightly constrained in the model fitting.

To test whether all the parameters of the models illustrated in Fig 3A were identifiable, we fitted each to simulated BrdU/Ki67 labelling timecourses that were generated by the model with added noise. For all models, the estimated total rate of influx of new cells into memory was strongly correlated with the net loss rate of the direct descendant(s) of the source,

indicating that these parameters are not separately identifiable. To resolve this, we obtained independent estimates of the rates of influx by modelling the slow accumulation of donor CD4 $T_{CM}$ and $T_{EM}$ within cohorts of busulfan chimeric mice over long timescales (Text C and Fig C of S1 File). The posterior distributions of these estimates (summarised in Table A of S1 File) yielded strong priors on the influx rates for the subsequent analyses.

We fitted the branched, linear, and burst models to each of the 8 timecourses—CD4 $T_{CM}$ and $T_{EM}$ separately for donor and host, in both young and old cohorts. All three models yielded excellent fits to the data; surprisingly, these fits were statistically and visually indistinguishable (Table B of S1 File). (Confirming our earlier conclusions [5], the simple temporal heterogeneity model in which Ki67$^{high}$ and Ki67$^{low}$ cells had different death and division rates performed poorly (Fig D of S1 File), and we do not consider it further here.) However, the parameters estimated for the linear and bursting model exhibited some colinearity (Fig E of S1 File, panels A and B), so we proceed by describing the branched model, whose parameters were most clearly identifiable (Fig E of S1 File, panel C). The fits of the branched model are overlaid in Fig 3C. The posterior distributions of its parameters are shown in Fig 4 and summarised in Table C of S1 File. Corresponding results for the linear and burst models are shown in Figs F and H and Table D of S1 File and Figs G and I and Table E of S1 File, respectively. Importantly, however, the conclusions we draw from here onwards are supported by all three models.

## Clonal age effects explain changes in CD4 $T_{CM}$ and $T_{EM}$ dynamics with mouse age

For both CD4 $T_{CM}$ and $T_{EM}$, the branched model fits indicated that in both young and older mice, cells that enter the faster subpopulation from the precursor pool divided and were lost over timescales of a few days. In contrast, slow cells divided rarely (mean interdivision times of 100 to 200 days) and had life spans of 60 to 100 days (Fig 4). While the differences in these two subpopulations are clear, these division and loss rate estimates are rather broad because they still exhibited some colinearity (Fig E of S1 File, panel C). We return to the issue of estimating memory cell lifetimes below. Quantities that were better defined, and perhaps immunologically more relevant, were the balance of loss (at rate $\delta$) and self-renewal (at rate $\alpha$) for each subpopulation. This net loss rate, $\lambda = \delta - \alpha$, defines the loss of a population in bulk, rather than the turnover of its constituent cells. For example, if the rate at which cells are lost is is balanced exactly by their rate of self-renewal, $\lambda = 0$ and the population is sustained indefinitely (that is, it is in dynamic equilibrium). In the context of our study, the time taken for a cohort of cells entering a population to halve in size is then $(\ln 2)/\lambda$. We estimated this quantity for the $T_{CM}$ and $T_{EM}$ populations in bulk, but this half-life is particularly relevant because it simultaneously defines the persistence of a TCR clone. To emphasise this point, we refer to $(\ln 2)/\lambda$ as a "clonal" half-life, even though we do not track individual clonotypes in our study. The similarity in the rates of division and loss of fast cells implied that fast CD4 $T_{CM}$ and $T_{EM}$ populations were remarkably persistent, with clonal half lives of between 35 and 70 days, and $T_{EM}$ slightly less persistent than $T_{CM}$. In contrast, the slower memory cells self-renewed rarely, so their clonal half-lives (70 to 140 days) were governed largely by the cell life span and were similar for $T_{CM}$ and $T_{EM}$.

Going beyond these broad characterisations, we wanted to identify the basis of the observed host/donor differences and any changes in memory dynamics with cell age or mouse age. In particular, in young mice, for both CD4 $T_{CM}$ and $T_{EM}$, we inferred that the higher expression of Ki67 within donor cells was due to a greater prevalence of fast cells, compared to host (Fig 4). In Text D of S1 File, we show that these differences arise because slow, host-derived memory cells were more persistent than their younger donor counterparts, while fast cells

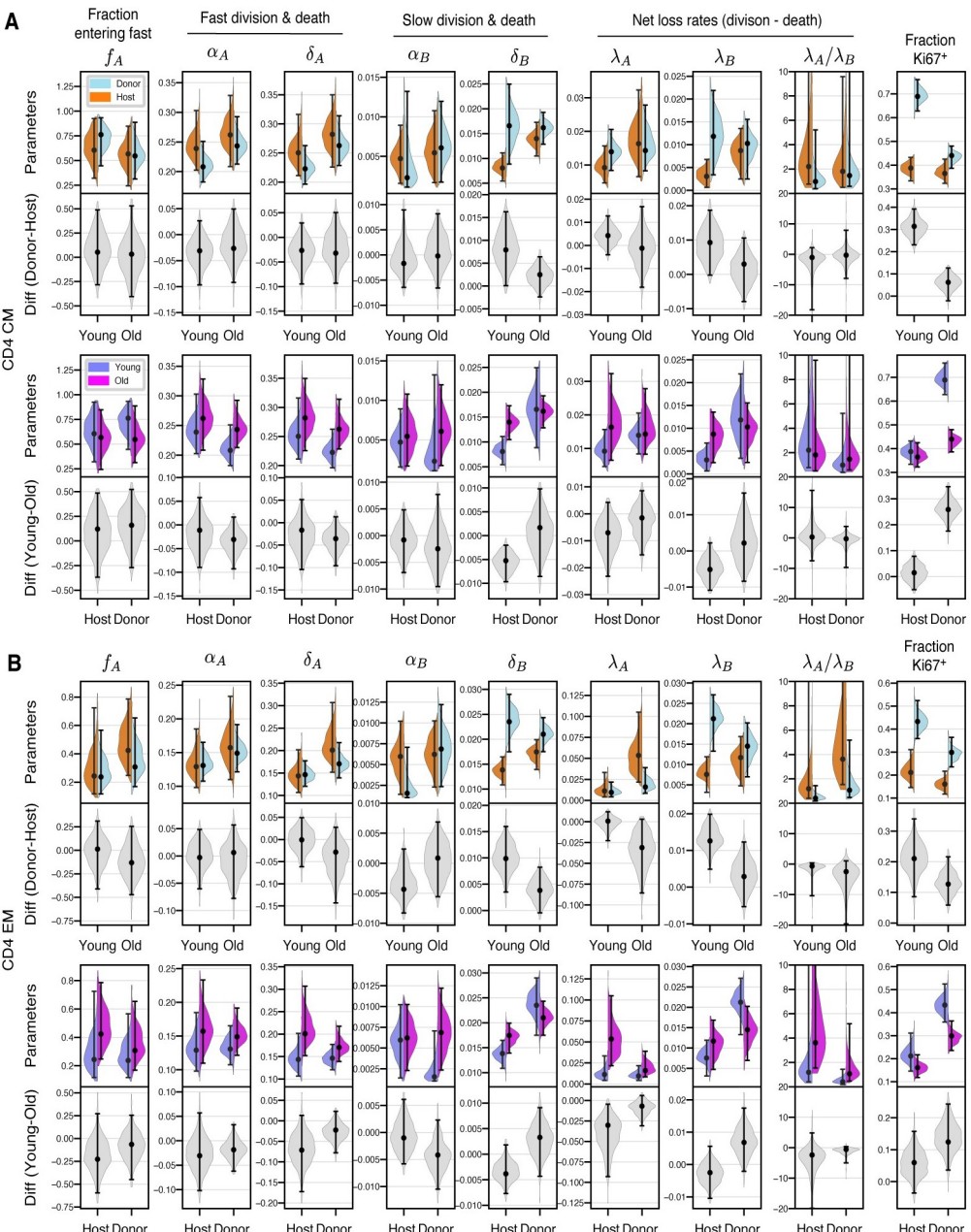

**Fig 4.** Posterior distributions of parameters derived from the branched model of the kinetics of CD4 $T_{CM}$ (**A**) and $T_{EM}$ (**B**), in the young and old cohorts of mice. Grey violin plots show differences in parameters (host/donor, and young/old). Points and bars show MAP estimates and 95% credible intervals, respectively. The data underlying this figure can be found in S1 Data.

behaved similarly, irrespective of cell age (host/donor status) or mouse age. We also found that this increase in the persistence of more quiescent memory clones with their age derived from increased cell life span, rather than an increase in rates of self renewal. These conclusions held for all three (branched, linear, and burst) models.

We expected the age structure of host-derived CD4 $T_{CM}$ and $T_{EM}$ to change between young and old cohorts to a lesser extent than that of the donor cells. Therefore, any changes linked

directly to mouse age ought to be more manifest among host cells. However, we saw no shifts in host cell kinetics between the age cohorts, reflected in the similarity of their BrdU/Ki67 labelling curves (Fig 3, panels C and D). Therefore, we could not detect any effects of mouse age on the dynamics of circulating memory CD4 T cells. We can also conclude that increases in the survival capacities of slow CD4 $T_{CM}$ and $T_{EM}$ with cell age must saturate over a timescale of weeks; if they did not, we would expect to see a decline in the loss rates of host-derived memory cells in older mice.

In summary, the stark differences in donor and host kinetics in younger mice are readouts of the effects of cell age, defined as the time since a cell or its ancestor entered the CD4 $T_{CM}$ or $T_{EM}$ population. As they age, slow CD4 $T_{CM}$ and $T_{EM}$ clones become more persistent, shifting the population as a whole to a more quiescent state and explaining the decline in Ki67 within these subsets as mice age. The convergent dynamics of host and donor cells in older mice reflect a convergence in their cell age profiles.

## Comparing memory CD4 T cell life spans to previous estimates

The other DNA labelling studies we are aware of that measured or inferred memory CD4 T cell life spans in mice reported the population average and did not distinguish $T_{CM}$ and $T_{EM}$ [4,8,34]. For comparison, we used 3 independent methods of deriving the corresponding mean life span from our data.

Westera and colleagues [4], using heavy water labelling, showed that it is necessary to allow for multiple subpopulations with different rates of turnover to obtain parameter estimates that are independent of the labelling period. They were not able to resolve the number of subpopulations or their rates of turnover individually but established a robust estimate of the mean life span of 15 days (range 11 to 15) in adult mice. Baliu-Pique and colleagues [8] employed the same approach to derive a life span of 8 days (95% CI 5 to 15). To obtain an analogous estimate, we used the model parameters to calculate the average loss rate of fast and slow cells among both donor and host, each weighted by their population sizes (Text E of S1 File). Fig 5A shows the posterior distributions of the life spans of $T_{CM}$ and $T_{EM}$ separately, with point estimates of roughly 8 and 21 days, respectively, and with little change with mouse age. Notably, our confidence in these quantities is greater than our confidence in the life spans of

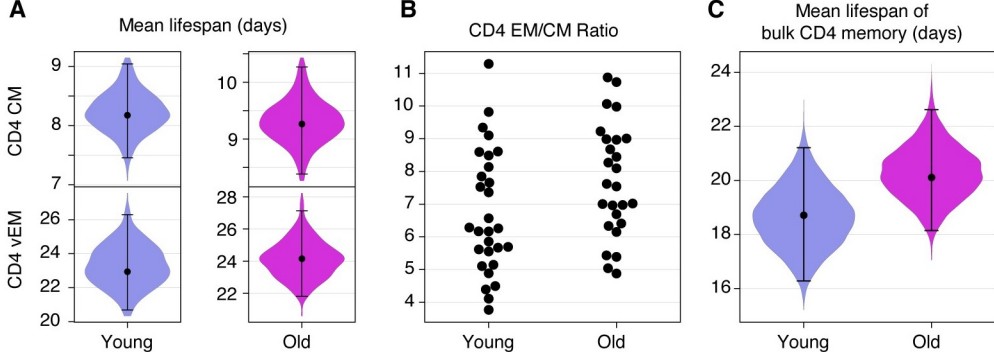

**Fig 5. Estimates of the mean life span of circulating memory CD4 T cells in the 2 age cohorts of mice. (A)** Expected life spans of CD4 $T_{CM}$ and $T_{EM}$ in young and old mice, each averaged over fast and slow subpopulations (see Text E of S1 File). **(B)** Relative abundance of lymph node-derived CD4 $T_{CM}$ and $T_{EM}$, by age. **(C)** Estimated mean life spans of memory CD4 T cells, averaging over $T_{CM}$ and $T_{EM}$, in the young and old cohorts. Violin plots show the distributions of the life spans over the joint posterior distribution of all model parameters; also indicated are the median and the 2.5 and 97.5 percentiles. The data underlying this figure can be found in S1 Data.

fast and slow populations themselves (Fig 4). To calculate a population-average life span, we account for the lower prevalence of $T_{CM}$ relative to $T_{EM}$ (Fig 5B). The resulting weighted estimates were 18 days (95% CrI 16 to 21) for younger mice, and 20 days (18 to 22) in the older cohort (Fig 5C).

We obtained estimates of the mean life span of memory CD4 T cells in 2 other ways. Both rely on estimating the total rate of production, which at steady state is balanced by the average loss rate, which is the inverse of the mean life span. One method is to use the initial upslope of the BrdU labelling curve, which reflects the total rate of production of new memory cells through division and influx. One can show that the mean life span is approximately $2\epsilon/p$, where $\epsilon$ is the efficiency of BrdU uptake and $p$ is the early rate of increase of the BrdU$^+$ fraction (Text F of S1 File). In the young cohort, these rates were roughly 0.08/day and 0.04/day for $T_{CM}$ and $T_{EM}$, respectively; in older mice, 0.1 and 0.05. Accounting again for the $T_{CM}$ and $T_{EM}$ abundances (Fig 5B), we obtain rough point estimates of the life span of 25 days in the younger cohort and 27 days in the older mice. A similar calculation was employed by De Boer and Perelson [34], who drew on BrdU labelling data from Younes and colleagues [17] to obtain a mean life span of between 14 and 22 days (Text F of S1 File).

The second method is to utilise Ki67 expression, which again reflects the production of cells through both influx and cell division. If the fraction of cells expressing Ki67 is $k$, and Ki67 is expressed for $T$ days after division, one can show that the mean life span is approximately $-T/\ln(1-k/2)$ (Text F of S1 File). The frequencies of Ki67 expression within CD4 $T_{CM}$ and $T_{EM}$ were roughly $0.4 \pm 0.1$ and $0.2 \pm 0.1$, respectively, and were similar in young and old mice (Fig 1B); using these values weighted by the abundances of $T_{CM}$ and $T_{EM}$ (Fig 5B, EM/CM $\simeq$ 7.5 $\pm$ 2), with our estimated $T$ = 3.1d gives a mean life span of approximately 25 days, although with some uncertainty (conservative range 15 to 70d).

In summary, we find robust estimates of the average life span of memory CD4 T cells that increase slightly with mouse age and are in line with estimates from other studies.

## Using patterns of chimerism to identify the precursors of CD4 $T_{CM}$ and $T_{EM}$

These models of CD4 $T_{CM}$ and $T_{EM}$ dynamics made no assumptions about the identity of their precursors—they only assumed that the rates of influx of host and donor precursors into each subset were constant over the course of the labelling assay and that new memory cells expressed Ki67 at high levels. Because the donor/host composition of this influx presumably reflects the chimerism of the precursor population, we reasoned that by comparing the chimerism of the flows into CD4 $T_{CM}$ and $T_{EM}$ to that of other populations, we could infer the lineage relationships between naive and circulating memory CD4 T cell subsets.

Fig 6 shows these chimerism estimates. Our analysis of the rate of accumulation of donor cells within CD4 $T_{CM}$ and $T_{EM}$ in busulfan chimeric mice (Text C and Fig C of S1 File) yielded the chimerism of the flows into these subsets, shown in green on the left of each panel in Fig 6. We also used the posterior distributions of parameters from the fitted models to estimate the chimerism within fast and slow subsets (split colour violin plots, shown for branched and linear models; those derived from the burst model were very similar, and for clarity are not shown). The points represent experimental observations.

As expected, chimerism in different cell populations showed considerably more variation in the young cohort, which had undergone BMT relatively recently, than in the older cohort in which donor/host ratios throughout the peripheral T compartments had had more time to equilibrate. We therefore focused on the younger cohort to establish potential differentiation pathways.

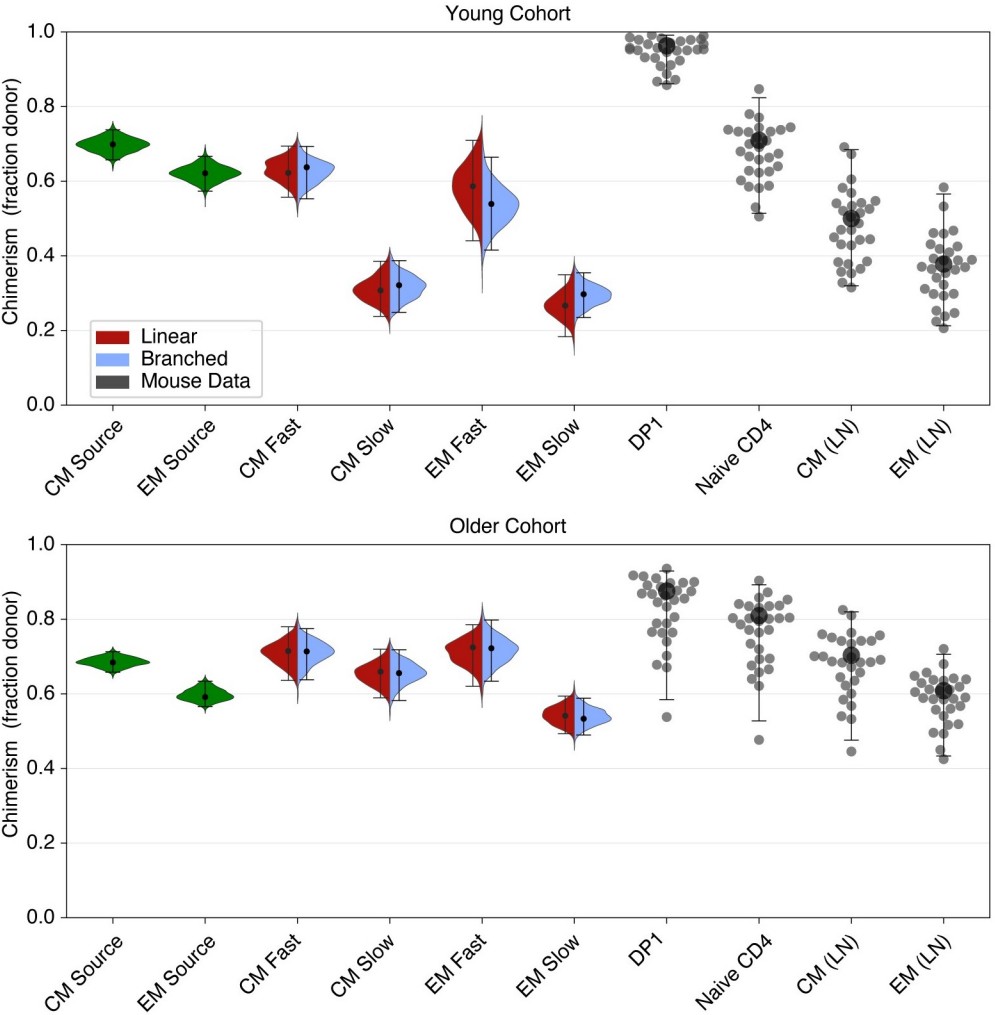

**Fig 6. Inferred and measured values of the chimerism (donor cell fraction) within thymic and peripheral T cell subpopulations.** Left-hand (green) violin plots indicate the chimerism of the constitutive influxes into CD4 $T_{CM}$ and $T_{EM}$ estimated using Eq 1 of Text C of S1 File, $f_d$. Split-colour violin plots indicate the posterior distributions of the chimerism within fast and slow subsets of CD4 $T_{CM}$ and $T_{EM}$; red and lilac indicate the linear and branched model predictions, respectively. Points represent experimentally observed donor fractions within early-stage double-positive thymocytes (DP1), naive T cells, and CD4 $T_{CM}$ and $T_{EM}$ derived from lymph nodes. The data underlying this figure can be found in S1 Data.

One source of new memory might be antigen-specific recent thymic emigrants (RTEs), whose chimerism would be expected to be similar to that of early-stage double positive thymocytes (DP1). However, the observed chimerism of DP1 cells (Fig 6) was much higher than the inferred chimerism of the $T_{CM}$ and $T_{EM}$ precursors, so our analyses do not support RTE as the dominant source of new CD4 MP T cells. Instead, the chimerism of $T_{CM}$ precursors aligned well with that observed within bulk naive CD4 T cells, and the chimerism of the $T_{EM}$ precursors aligned with that of $T_{CM}$. This pattern clearly suggests a naive → $T_{CM}$ → $T_{EM}$ developmental pathway for CD4 MP T cells, consistent with the order in which donor-derived cells appear within these subsets in busulfan chimeras (Fig C of S1 File). It was also clear that the estimated chimerism within slow memory cells was too low for them to be an exclusive source for any other population, consistent with their being predominantly a terminal differentiation stage in

memory T cell development. Instead, our observations are consistent with a model of development in which new CD4 $T_{EM}$ derive predominantly from more rapidly dividing CD4 $T_{CM}$.

### Age-dependent survival rates explain shortfalls in replacement within $T_{CM}$ and $T_{EM}$

In older busulfan chimeric mice, the replacement curves for CD4 $T_{CM}$ and $T_{EM}$ saturated at donor fractions of approximately 0.7 and 0.6, respectively (Fig 6 and Fig C of S1 File), lower than that of naive CD4 T cells (Fig 6). Given that the average life spans of CD4 $T_{CM}$ and $T_{EM}$ are a few weeks, it is then perhaps puzzling that the donor fractions within all three compartments do not equalise at late times after BMT. Our analysis suggests that these shortfalls are an effect of cell age–dependent survival. The mean life span is a rather crude measure of memory turnover, as life spans are significantly right-skewed; in addition, slower cells increase their life expectancies as they age. This increase will lead to a first-in, last-out effect in which donor T cell clones will always be at a survival disadvantage, on average, relative to their older host counterparts. This effect naturally leads to a shortfall in the donor fraction within each memory subset relative to its precursor (putatively, $T_{CM}$ relative to naive; and $T_{EM}$ relative to $T_{CM}$). As the mice age and the donor and host cell age distributions converge, this shortfall decreases but still remains (Fig 6).

### Validations of model assumptions and fits in other systems

Previously, we showed that the total numbers of memory CD4 T cells are indistinguishable in busulfan chimeric mice and wild-type (WT) controls [5]. However, there remains the possibility that the observed differences in the behaviour of host and donor memory CD4 T cells in busulfan chimeras derives in some way from the impact of drug treatment and HSC transplant, rather than simply from their different age profiles. We therefore sought to use 2 independent mouse reporter systems to validate the structure of our models, and the results of the model fitting.

First, we used an adoptive transfer approach to test the assumptions underlying our models. $Ki67^{mCherry-CreERT}$ $Rosa26^{RcagYFP}$ mice express a Ki67-mCherry fusion protein and inducible CreERT from the $Mki67$ locus [26] together with a $Rosa26^{RcagYFP}$ Cre reporter construct (Materials and methods). Treatment of donor mice with tamoxifen therefore induces YFP in cells expressing Ki67. Three days following treatment, we isolated either bulk CD4⁺ conventional T cells or purified CD4⁺ $T_{EM}$ from the lymph nodes of these reporter mice and adoptively transferred them into WT CD45.1 congenic hosts (Fig J of S1 File, panel A). As expected, Ki67 levels within YFP⁺ cells were higher than among YFP⁻ cells (Fig 7A and Fig J of S1 File, panel B); having divided in the days prior to transfer, the YFP⁺ cells were enriched for faster-dividing populations. We then assessed Ki67 expression among the transferred cells 7 days after transfer (Fig 7B and Fig J of S1 File, panel C). We made two key observations. First, we found that Ki67 levels within donor-derived CD4 $T_{EM}$ declined when they were transferred alone but were preserved following the transfer of CD4 T cells in bulk. This result is consistent with our basic assumption of continued production of new, Ki67^high CD4 $T_{EM}$ from a precursor that is within circulating CD4 T cells. Second, we observed that Ki67 levels among YFP⁺ cells also fell and converged with that of YFP⁻ cells. The decline is consistent with the assumption of kinetic heterogeneity; if memory cells divided and died at a constant rate, we would expect the level of Ki67 within the YFP⁺ cells to be preserved. Notably, the decline and convergence of Ki67 levels within YFP⁺ and YFP⁻ cells is predicted by all three models (Text G of S1 File).

Second, we used $CD4^{CreERT}$ $Rosa26^{RmTom}$ mice to directly test the predictions of our fitted models. In this system, the Cre reporter is constructed such that cells expressing CD4 during tamoxifen treatment permanently and heritably express the fluorescent reporter mTomato

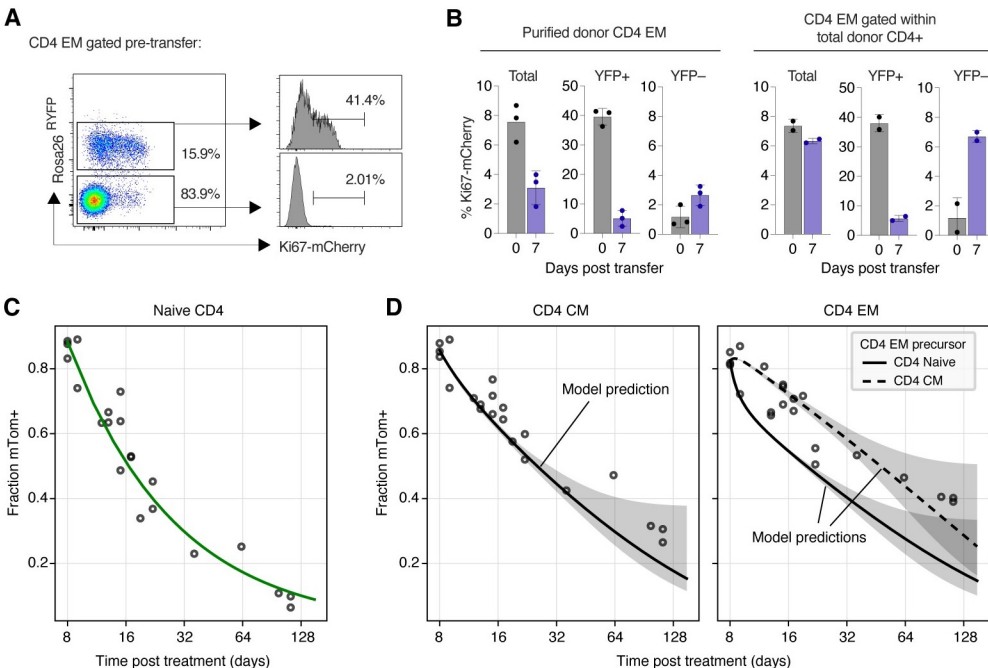

**Fig 7. Validation of the structure and predictions of the models of memory CD4 T cell dynamics. (A)** CD4 $T_{EM}$ after tamoxifen treatment, pretransfer, with Ki67 expression stratified by YFP expression. **(B)** Proportions of cells in bulk, YFP$^+$ and YFP$^-$ CD4 $T_{EM}$ expressing Ki67 at days 0 and 7 after transfer. **(C)** Empirical description (fitted green curve) of the observed timecourse of the frequency of mTom$^+$ cells within the naive CD4 T cell pool, after tamoxifen treatment of CD4 reporter mice. Dilution is driven by the influx of unlabelled cells from the thymus. **(D)** Observed and predicted frequencies of mTom$^+$ cells within the CD4 $T_{CM}$ and $T_{EM}$ following tamoxifen treatment. Shaded regions span the 2.5 and 97.5 percentiles of the distribution of predicted trajectories, generated by sampling 1,000 sets of parameters from their joint posterior distribution. The data underlying this figure can be found in S1 Data.

(mTom). In a closed self-renewing population, the frequency of cells expressing mTom will therefore remain constant. Any decline in this frequency therefore reflects the influx of unlabelled precursors. We tamoxifen-treated a cohort of CD4 reporter mice that were age-matched to our younger cohort of busulfan chimeras and followed them for approximately 18 weeks (Fig J of S1 File, panel D). We described the timecourse of the mTom$^+$ fraction within naive CD4 T cells empirically (Fig 7C, green curve). We then used the parameters of our fitted models to predict the timecourses of mTom$^+$ frequencies within CD4 $T_{CM}$ and $T_{EM}$ (Fig 7D; see Text H of S1 File for modelling details). The predictions of all three models were identical, and for both $T_{CM}$ and $T_{EM}$ agreed well with the data, with better visual support for $T_{CM}$ as a precursor to $T_{EM}$ (Fig 7D, rightmost panel). This agreement demonstrates that the kinetics of memory CD4 T cells derived from busulfan chimeric mice are indistinguishable from those in mice that have not undergone HSC depletion and replacement, and so supports our assertion that differences in behaviour of host and donor T cells derive only from their different age distributions and not any ontogenic differences. The simulations also lend support to a dominant CD4 $T_{CM} \rightarrow T_{EM}$ differentiation pathway.

## Discussion

Our key result is that the longer a cohort of CD4 $T_{CM}$ or $T_{EM}$ persists in memory, the greater the expected life spans of its constituent cells. A consequence of this effect is that the average

loss rate of antigen-specific memory CD4 populations decreases as they age. This behaviour mirrors that of naive CD4 and CD8 T cells that we and others have identified in both mice [27,28,36] and humans [37], suggesting it is an intrinsic property of T cells.

This dynamic will lead to the accumulation of older memory CD4 T cells. In contrast, a recent study argued that an increase in cell mortality with cell age may act to preserve the TCR diversity of memory populations as an individual ages, and, hence, be beneficial [38]. In the latter study, cell aging was directly associated with cell divisions, whereas here we defined age to be a heritable clock that measures the time since a cell's ancestor entered the memory pool. These two measures of age are positively correlated, but any division-aging effects would manifest more strongly among fast memory cells than among slow. Indeed, we found that slow memory cells self-renew rarely. It has also been shown that memory cells deriving from aged naive T cells may be functionally impaired [39–41]. It is therefore unclear what cell age effects are optimal for the long-term preservation of functional memory repertoires.

The mechanistic basis of any change in survival with cell age is unclear. Memory T cells may undergo maturation that increases their expected time-to-die as they age, such as the gradual acquisition of senescence markers. An alternative is a simple selection effect in which every cohort of new memory T cells is heterogeneous in its survival capacity, and more persistent clones are selected for over time. Another possibility is that memory cells experience slow changes in their microenvironments; for example, progressive alterations in receptor expression that might alter their circulation patterns and influence the availability of survival signals. This shift in trafficking patterns would then be a proxy for cell aging. To explore this idea, one would need to perform histology to study the spatial distributions of donor- and host-derived memory cells in the younger cohorts of mice, where the average ages of the two populations are most different.

The development and long-term maintenance of memory CD4 T cell subsets has been studied less extensively than those of CD8 T cells, but the dynamics of the latter have some key similarities with the results we describe here and may help to resolve some of the ambiguities in our analyses. A study involving deuterium labelling of yellow fever virus (YFV)-specific CD8 T cells in humans [42] demonstrated a progressive shift to quiescence over months to years, with surviving memory T cells being very long lived and dividing rarely. Zarnitsyna and colleagues [43] explored a variety of mathematical models to quantify this kinetic and found the strongest support for rates of both division and death that decline with time and result in a power-law decrease in memory cell numbers. Akondy and colleagues [42] found that cells persisting long term were deuterium-rich, indicating that they were derived from populations that divided extensively in the days to weeks following challenge. This kinetic gives support to the linear or burst models of memory, rather than the branched model in which the more persistent cells are quiescent from the moment they enter the memory population.

MP CD4 T cells divide more rapidly on average than conventional antigen-specific memory cells [15–17]. Younes and colleagues [17] argued that MP CD4 T cells comprise a proliferative subset and a slower population with the properties of conventional memory. This distinction was motivated by the observation that bulk and LCMV-specific memory CD4 T cells, identified with tetramer staining 15 days after LCMV challenge, were phenotypically similar but had Ki67 expression levels of 25% to 40% and 5%, respectively. Bulk memory is presumably enriched for MP cells, and indeed, the frequencies of Ki67 expressing cells they observed among the tetramer-negative population are consistent with the roughly 30% weighted average among CD4 $T_{CM}$ and $T_{EM}$ in our analyses (Fig 2B).

There have been conflicting reports regarding the signals required for the generation and maintenance of fast MP CD4 T cells. Younes and colleagues argued that they are driven by cytokines, based on the lack of an effect of an MHC class II-blocking antibody and a modest

reduction in proliferation under IL-7R blockade [17]. Their results contrasted with reports that rapidly dividing MP CD4 T cells depend on interactions with MHC class II and require proximal TCR signalling kinases to sustain proliferation [16,44] and have reduced dependence on IL-7 compared to antigen-specific CD4 memory [15]. However, later studies from the same lab as Younes and colleagues identified the importance of costimulation signals from CD28 for the proliferation of fast cells even though division was not affected by MHC class II blockade [12]. The same study highlighted the importance of TCR and CD28 costimulation signals for the generation of MP CD4 T cells from naive precursors. These observations can be reconciled if the generation and maintenance of MP CD4 T cells rely on CD28 and TCR interactions but vary in the threshold of TCR signalling required and if there is redundancy with CD28 signals. In addition, interpretation of MHC class II blockade experiments is heavily dependent upon whether antibody blockade is considered absolute, or, as is more likely the case, only partial. If the latter is true, the reduction in availability of MHC class II ligands may be sufficient to block priming events but insufficient to halt self-renewal of existing memory.

Given the involvement of TCR signals in the production of new MP CD4 T cells, the difference in average levels of proliferation within bulk MP CD4 T cells and antigen-specific memory observed by Younes and colleagues [17] might be explained quite simply by the fact that MP T cells are continually produced in response to chronic stimuli with self or commensal antigens, while LCMV-specific memory will become less proliferative with time post-challenge as the acute infection resolves. The transfer experiment we described here supports this interpretation; CD4 $T_{EM}$ in isolation lost Ki67 expression, whereas when CD4 T cells were transferred in bulk, Ki67 levels on CD4 $T_{EM}$ were maintained, presumably because their precursors were also present and continued to be stimulated. Similar reasoning might explain the report by Choo and colleagues that memory CD8 T cells specific for epitopes of LCMV divide at the same rate as memory CD8 T cells in bulk [45]. They established this equivalence by labelling each population with CFSE, transferring them into congenic recipient mice and comparing their CFSE dilution profiles; by doing so, the source of any new, more proliferative MP CD8 cells was presumably removed or diminished, and the remaining polyclonal populations divided at the same slow rate as the LCMV-specific cells. In summary, MP T cells, rather than being distinct from conventional antigen-specific memory cells, may in fact represent a cross-section of the entire process of memory generation and maintenance.

The binary characterisation of fast and slow that we model here may be a simple representation of a more heterogeneous system. Younes and colleagues [17] saw a 4-fold disparity in Ki67 expression between tetramer-positive (LCMV-specific) and tetramer-negative (MP) CD4 T cells 15 days post-challenge, indicating that fast LCMV-specific memory persisted for only a few days. This transience contrasts to the more extended residence times of fast populations, at the clonal level, that we inferred here. The MP populations we studied likely respond to a variety of commensal or environmental antigens. CD4 $T_{CM}$ and $T_{EM}$ may therefore exhibit variability in their persistence times in a clone-specific way, perhaps depending on antigenic burden, TCR specificity, or their T helper phenotype [46].

By augmenting measurements of BrdU content with Ki67 expression, here and in a previous study, we could reject a model of "temporal heterogeneity" in which cells expressing Ki67 are both more likely to divide again and at increased risk of loss or onward differentiation. However, the burst model we considered here is a generalisation in which the duration of this more "risky" post-division state is no longer tied to expression of Ki67 and is instead a free parameter. We estimated the transition rate from fast to slow to be quite low, such that the average time spent in the bursting state is determined instead by the cell life span, which was 4 to 5 days. Consequently, the efficiency of return to quiescence from the burst phase was rather

low, with only approximately 1% of cells surviving the transition. We found a comparably low efficiency of survival from fast to slow in the linear model. With this additional flexibility, the burst model yielded essentially identical fits to the branched and linear models. We also found that all three models yielded similar predictions for the outcome of the experiment that followed the behaviour of cohorts of cells enriched for fast or slow cells (Fig 7). Therefore, the kinetic substructures of CD4 $T_{CM}$ and $T_{EM}$ remain unclear. We speculate they may only be decipherable with single-cell approaches such as barcoding.

Our analyses gave clues as to the sources of new CD4 $T_{CM}$ and $T_{EM}$. By comparing the donor cell content of the constant flow of new cells into the CD4 $T_{CM}$ and $T_{EM}$ pools to the donor content of other cell subsets, we saw clear evidence for a flux from $T_{CM}$ to $T_{EM}$, likely from the rapidly dividing $T_{CM}$ population. We also found evidence that newly produced $T_{CM}$ derive largely from the naive CD4 T cell population in bulk. This was perhaps surprising; among mice housed in clean conditions, without overt infections, we expect naive T cells with appropriate TCR specificities for self or commensal antigens will be recruited into memory efficiently. In that case, one might expect recent thymic emigrants to be the dominant source of new memory, and so for this influx to be highly enriched for donor T cells. Instead, we saw continued recruitment of host cells into memory long after BMT, suggesting that even within SPF housing conditions, mice are regularly exposed to new antigenic stimuli. Alternatively, there may be a stochastic component to recruitment, such that naive T cells of any age have the potential to be stimulated to acquire a memory phenotype. Support for the latter idea comes from studies that implicate self antigens as important drivers of conversion to memory [12,18], and that proliferation is greatest among CD5hi cells [12], a marker whose expression correlates with TCR avidity for self. Conversion to memory is therefore likely be an inefficient process since we would expect the naive precursors to be relatively low affinity for self-antigens; the majority of T cells with high affinity for self are deleted in the thymus. The efficiency of recruitment could also be subject to repression by regulatory T cells. Further, as mentioned above, within these relatively low affinity memory populations, any dependence of survival on affinity for self may lead to selection for fitter clones with cell age and may explain the increase in the average clonal life expectancy over time. The preferential retention of such clones may then be relevant to understanding the increasing incidence of autoimmune disease that occurs with old age. These ideas need to be tested in future studies.

## Material and methods

### Generation of busulfan chimeric mice

Busulfan chimeric C57Bl6/J mice were generated as described in [24]. In summary, WT CD45.1 and CD45.2 mice were bred and maintained in conventional pathogen-free colonies at the Royal Free Campus of University College London. CD45.1 mice aged 8 weeks were treated with 20 mg/kg busulfan (Busilvex, Pierre Fabre) to deplete HSC and reconstituted with T cell–depleted bone marrow cells from congenic donor WT CD45.2 mice. Chimeras were killed weeks after bone marrow transplantation. Cervical, brachial, axillary, inguinal, and mesenteric lymph nodes were dissected from mice; single-cell suspensions prepared and analysed by flow cytometry. BrdU (Sigma) was administered to busulfan chimeric mice by an initial intraperitoneal injection of 0.8 mg BrdU, followed by maintenance of 0.8 mg/mL BrdU in drinking water for the indicated time periods up to 21 days. BrdU in drinking water was refreshed every 2 to 3 days. All experiments were performed in accordance with UK Home Office regulations, project license number PP2330953.

### Reporter mouse strains

Ki67<sup>mCheery-CreERT</sup> reporter mice (*Mki67<sup>tm1.1(cre/ERT2)Bsed</sup>*) have been described previously [26]. These mice were crossed with a Cre reporter strain in which a CAG promoter driven YFP is expressed from the *Rosa26* locus following Cre mediated excision of upstream transcriptional stop sequences; B6.Cg-Gt(ROSA)26<sup>Sortm3(CAG- EYFP)Hze/J</sup> (Jax strain 007903). Ki67<sup>mCherry-CreERT</sup> Rosa26<sup>RcagYFP</sup> mice were homozygous for the indicated mutations at both loci. CD4<sup>CreERT</sup> Rosa26<sup>RmTom</sup> reporter mice were generated by breeding Cd4<sup>CreERT2</sup> mice [47] with B6.Cg-Gt(ROSA)26<sup>Sortm9(CAG-tdTomato)Hze/J</sup> mice (Jax strain 7909). Experimental mice were heterozygous for the indicated mutations at both loci. Mice were treated with tamoxifen by a single intraperitoneal injection with 2 mg of tamoxifen (Sigma) diluted in corn oil (Fisher Scientific).

### Flow cytometry

Cells were stained with the following monoclonal antibodies and cell dyes: CD45.1 FITC, CD45.2 AlexaFluor 700, TCR-beta APC, CD4 PerCP-eFluor710, CD25 PE, CD44 APC-eFluor780, CD25 eFluor450, CD62L eFluor450 (all eBioscience), TCR-beta PerCP-Cy5.5, CD5 BV510, CD4 BV650, CD44 BV785 (all BioLegend), CD62L BUV737 (BD Biosciences), LIVE/DEAD nearIR and LIVE/DEAD Blue viability dyes (Invitrogen). BrdU and Ki67 costaining was performed using the FITC BrdU Flow Kit (BD Biosciences) according to the manufacturer's instructions, along with anti-Ki67 eFluor660 (eBioscience). Cells were acquired on a BD LSR-II or BD LSR-Fortessa flow cytometer and analysed using Flowjo software (Treestar). Subset gates were as follows: CD4 naive: live TCR$\beta^+$ CD5$^+$ CD4$^+$ CD25$^-$ CD44$^-$ CD62L$^+$. CD4 T$_{EM}$: live TCR$\beta^+$ CD5$^+$ CD4$^+$ CD25$^-$ CD44$^+$ CD62L$^-$. CD4 T$_{CM}$: live TCR$\beta^+$ CD5$^+$ CD4$^+$ CD25$^-$ CD44$^+$ CD62L$^+$.

### Cell transfers

For adoptive transfers, donor Ki67<sup>mCherry-CreERT</sup> Rosa26<sup>RcagYFP</sup> mice were injected with a single dose of tamoxifen, and three days later, total lymph nodes were isolated and CD4 T cells enriched using EasySep Mouse CD4$^+$ T cell isolation kit. Cells were then labelled for CD25, CD4, TCR, CD62L, and CD44, and total CD4$^+$ conventional or CD4 EM further purified by cell sorting on a BD FACS Aria Fusion. Cells ($10^6$/mouse) were transferred by tail vein injection into CD45.1 WT hosts. Seven days later, hosts were culled and donor cell phenotype in host lymph nodes analysed by flow.

### Mathematical modelling and statistical analysis

The mathematical models and our approach to model fitting are detailed in Text A and Text B of S1 File, respectively. All code and data used to perform model fitting, and details of the prior distributions for parameters, are available at https://github.com/elisebullock/PLOS2024 and also at doi: 10.5281/zenodo.11476381. Models were ranked using the Leave-One-Out (LOO) cross validation method [48]. We quantified the relative support for models with the expected log pointwise predictive density (ELPD), for which we report the point estimate with standard error. Models with ELPD differences below 4 were considered indistinguishable.

### Supporting information

**S1 File. All supporting figures, tables, and text.**
(PDF)

**S1 Data. The data underlying the figures throughout the text and Supporting information.**
(XLSX)

## Author Contributions

**Conceptualization:** Andrew J. Yates, Benedict Seddon.

**Data curation:** M. Elise Bullock, Thea Hogan, Cayman Williams, Minahil Sharjeel.

**Formal analysis:** M. Elise Bullock, Cayman Williams, Sinead Morris, Maria Nowicka, Christiaan van Dorp.

**Funding acquisition:** Andrew J. Yates, Benedict Seddon.

**Investigation:** M. Elise Bullock, Thea Hogan, Cayman Williams, Maria Nowicka, Minahil Sharjeel, Christiaan van Dorp, Andrew J. Yates, Benedict Seddon.

**Methodology:** M. Elise Bullock, Thea Hogan, Christiaan van Dorp, Andrew J. Yates, Benedict Seddon.

**Project administration:** Andrew J. Yates, Benedict Seddon.

**Software:** M. Elise Bullock, Sinead Morris, Maria Nowicka.

**Supervision:** Andrew J. Yates, Benedict Seddon.

**Visualization:** M. Elise Bullock, Andrew J. Yates.

**Writing – original draft:** M. Elise Bullock, Andrew J. Yates, Benedict Seddon.

**Writing – review & editing:** M. Elise Bullock, Andrew J. Yates, Benedict Seddon.

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
