## [Editor Report · Decision Letter 0]

15 Oct 2023

Dear Dr. Yates, 

Thank you for submitting your manuscript entitled "Cell age, but not chronological age, governs the dynamics and longevity of circulating CD4 + memory T cells" for consideration as a Research Article by PLOS Biology.

Your manuscript has now been evaluated by the PLOS Biology editorial staff and I am writing to let you know that we would like to send your submission out for external peer review as an Update Article. Please select this option in the system when re-submitting. 

Once your full submission is complete, your paper will undergo a series of checks in preparation for peer review. After your manuscript has passed the checks it will be sent out for review. To provide the metadata for your submission, please Login to Editorial Manager (https://www.editorialmanager.com/pbiology) within two working days, i.e. by Oct 17 2023 11:59PM.

Kind regards,

Paula

---

Senior Editor

PLOS Biology

---

## [Decision Letter · Decision Letter 1]

20 Dec 2023

Dear Dr Yates,

Thank you for your patience while your manuscript entitled "Cell age, not chronological age, governs the dynamics and longevity of circulating CD4+ memory T cells" was peer-reviewed at PLOS Biology as an Update Article. Please also accept my sincere apologies for the delay in providing you with our decision. The manuscript has now been evaluated by the PLOS Biology editors, an Academic Editor with relevant expertise, and by three independent reviewers. 

As you will see, the reviewers find the conclusions interesting, but they also raise several issues that would need to be addressed before we can consider the manuscript for publication. Reviewer 1 has concerns about the model on whether you are comparing different processes between host and donor cells – older host cells with younger donor cells, rather that memory with naïve T cells – and also regarding the cell-age effects vs cell environment. In addition, this reviewer thinks you should discuss better the likely composition of the cells referred as ‘memory’ cells, among other issues. Reviewer 2 asks for additional analyses, including checking if CD4 and CD8 memory cells show a similar behaviour and several clarifications. Reviewer 3 thinks that the limitations should be expressed more clearly, so we suggest adding a section on limitations of the study to the discussion to address this point.

In light of the reviews, we would like to invite you to revise the work to thoroughly address the reviewers' reports. Given the extent of revision needed, we cannot make a decision about publication until we have seen the revised manuscript and your response to the reviewers' comments. Your revised manuscript is likely to be sent for further evaluation by all or a subset of the reviewers.

**IMPORTANT - SUBMITTING YOUR REVISION**

3. Resubmission Checklist

a) *PLOS Data Policy*

b) *Published Peer Review*

d) *Blurb*

Please also provide a blurb which (if accepted) will be included in our weekly and monthly Electronic Table of Contents, sent out to readers of PLOS Biology, and may be used to promote your article in social media. The blurb should be about 30-40 words long and is subject to editorial changes. It should, without exaggeration, entice people to read your manuscript. It should not be redundant with the title and should not contain acronyms or abbreviations. For examples, view our author guidelines: https://journals.plos.org/plosbiology/s/revising-your-manuscript#loc-blurb

Sincerely,

Ines

--

Ines Alvarez-Garcia, PhD

Senior Editor

PLOS Biology

Reviewers' comments

Rev. 1:

In this article, Bullock et al have combined wet lab experiments with mathematical modelling to examine memory phenotype CD4 T cells in mice. The field struggles to classify these cells that likely are composed of different populations that have responded following stimulation with cytokines and/or antigens, and in some cases are specific for antigens which are presented for short periods of time or specific for antigens which persist. The authors dissect the population of memory CD4 T cells in various ways - first by whether they express the lymph node homing molecule, CD62L and are classified by the field as Tcentral memory (CD62L+) or Teffector memory (CD62Llo/negative). Second, by whether they express a congenic molecule that marks them as host derived, or derived from a transfer of bone marrow stem cells when the hosts are around 50-100 days old. The authors also follow on from previous findings in the field describing that some memory phenotype CD4 T cells proliferate frequency and some very infrequently and there is not currently a consensus on an underlying mechanism to explain these different behaviours.

The authors' aims were to understand the proliferation of the CD4 T cells within these populations, predict life spans, and investigate the relationship between the memory populations.

The main findings described in the manuscript are:

1. Fast and slow dividing memory CD4 T cells can be found in both Tcm and Tem populations and in young and older mice.

2. But memory cells are more likely to divide in younger than older mice and at both time points, the donor cells are more likely to divide than the original host memory cells.

3. The differences in proliferation kinetics are less pronounced in the older mice than the younger mice.

4. The formation of Tem and Tcm cells can be explained by two different models in which Tcm and Tem are separate populations (branching) or in which Tcm convert into Tem (linear).

5. The life span of memory cell populations is similar to previous calculated values from other investigators.

The authors use these findings to conclude that cell age, rather than host environment, predict the proliferate behaviour of memory phenotype CD4 T cells. These data are potentially important as the longevity of memory CD4 T cells is key to understanding protective immunity to pathogens that the host as met previously through infection or vaccination. Long lived populations of CD4 T cells are more likely to provide durable immunity than short lived cells.

Overall, this is a well-written manuscript that combines the strengths of the authors in wet lab studies and mathematical modelling. I am a wet lab biologists and have limited ability to review the veracity of the modelling aspects of the manuscript.

The main strengths are that the authors have generated a substantial data set that allows for analysis of populations of memory CD4 T cells across time and which can probe the proliferation dynamics of the cells and generated different models to test hypotheses about the relationships between the cells.

The main weaknesses are that the authors have made a number of assumptions as they moved from findings to conclusions (see points 1 and 2 below) and that much of the work confirms previous findings (lifespan of memory populations) rather than presenting novel conclusions or hypotheses (i.e. differentiation of naïve cells into memory cells that may/may not require cells to go through a Tcm before a Tem phenotype).

Main concerns:

1. Are the authors modelling different processes between host and donor cells?

The diagram in Fig1A suggests that naïve CD4 T cells are predominantly donor derived at the analysis time point; the data in Fig2D suggests that donor derived memory phenotype cells are more likely to be Ki67+ than host memory cells. Is the explanation for the latter that the cells are coming from the naïve population? The authors investigate this later in the manuscript and conclude that, at least the Tcm population, is likely derived from the naïve CD4 T cell pool. They state in the discussion (line 419) that there is 'substantial recruitment of host cells into memory' and imply these are coming from the naïve pool. It wasn't clear to me what data this referred to and how the author ruled out that the host proliferating memory cells were memory cells re-activated by antigen and/or cytokine while the donor cells where newly activated naïve cells entering the activated/memory pool for the first time.

Thus, it is not clear to me how the authors can argue that they are comparing 'older' host cells with 'younger' donor cells rather than 'memory' with 'naïve' T cells. If they can't make this distinction, their main conclusion about the distinctions in proliferative behaviour between young and old cells is not supported. In the older cohort of mice, the differences between the host and donor cells is much less - this may be because here they are mainly comparing memory host cells with memory donor cells.

2. Argument that 'cell-age' effects dominate over the cell's environment.

This follows on from the concern above but contains a further distinction that the authors have not considered. This is that while the donor and host CD4 T cells are within the same host, they may be found in distinct microenvironments in the secondary lymphoid organs, and thus exposed to different levels of cytokines, different types of antigen presenting cells, and thus more or less likely to be exposed to antigen.

3. Lifespans - the authors describe half-lives (35-140 days) and mean lifespan (18-20 days). Can they explain the different measurements?

4. The authors have limited discussion on the cells that included/excluded from the populations. They have excluded CD25+ cells and thus will be excluding some, but not all of the Tregs. While CD25 exclusion will also mean they are excluding recently activated T cells from the analysis, CD25 is very transient on activated cells and thus their 'memory' populations are likely to include 'activated' cells. The authors themselves state that memory phenotype cells are derived from naïve cells and that some of these may be responding to environmental antigens even without overt infection/vaccination. It would be helpful if the authors include a few statements to describe the likely composition of the cells they refer to as 'memory' cells.

Minor concerns:

1. In the abstract, the author refer to clones - they have examined populations of cells not individual clones so this language is perhaps not appropriate.

2. It would be useful if the authors provide example gating for the T cells analysed, e.g. in a supplemental figure. It is also now standard to include the fluorochrome used in the axis label of the flow plot e.g. in Fig 1C and to include the axes numbers.

3. In Figure 1 part C Ki67/BrdU panels, it is unclear if the cells displayed are all T cells or the CD45.1 or CD45.2 cells. The legend suggests that are 'stratified' by donor/host but it wasn't clear which (or both) are shown.

4. In the methods, the authors should state the age of the mice used for the donor bone marrow, are these the same age as the host animals? This is important and HSC can show effects of age.

5. In Fig3C, CD4 Tcm proportion graph, it looks like there is only one blue/donor point at day 15. I am guessing there two points are on top of each other? Can they nudge one point slightly to enable the reader to see both points.

6. In the legend of Fig 3, it would be helpful to state that the lines show the predicted curves for the branched model, this is my understanding from the text in the paragraph with lines number 185-194.

7. Unclear what the word 'cohoxhich' means on line 295. Perhaps meant to say 'cohort in which'?

8. Extra 'was' in line 390.

9. The methods should contain information on which lymph nodes were examined and how the lymph nodes were processed and counted.

10. The methods should state how the busulfan was given to the mice, how many donor cells were transferred, how the donor cells were transferred and how the bone marrow was prepared (especially how depleted of B and T cells). Alternatively, If a prior publication contains detailed methods, this could be referenced in the methods section.

In terms of scope for an 'Update Article' my understanding is that the manuscript follows on from Rane et al, PLOS Biology, 2018. The 2018 paper examines survival of naïve CD4 T cells while the new manuscript examines memory CD4 T cells. The two studies thus ask similar questions of distinct populations using some similar methods. Thus, the two studies are certainly related but the results do go beyond the findings in the 2018 paper.

Rev. 2: Jose Borghans – note that this reviewer has signed his review

This paper elegantly tears apart the effects of the age of the host versus the age of the cell on the dynamics of circulating memory CD4+ T cells, and thereby gives important insights into the long-term maintenance of immunological memory. Generally speaking, the manuscript is clearly written, makes an important contribution to the field of immunological memory, is based on robust analyses, and in my opinion fits the scope of PLoS Biology.

I have a few questions and suggestions:

- I was wondering why the authors "only" report results for CD4 memory T cells. I would assume one could obtain information from both CD4 and CD8 memory T cells

---

## [Decision Letter · Decision Letter 2]

28 Mar 2024

Dear Dr Yates,

Thank you for your patience while we considered your revised manuscript "Cell age, not chronological age, governs the dynamics and longevity of circulating CD4+ memory T cells" for publication as a Update Article at PLOS Biology. This revised version of your manuscript has been evaluated by the PLOS Biology editors, the Academic Editor and the original reviewers.

Based on the reviews and on our Academic Editor's assessment of your revision, we are likely to accept this manuscript for publication, provided you satisfactorily address the remaining points raised by the reviewers. Please also make sure to address the following data and other policy-related requests.

* We would like to suggest a different title to improve readability: "The dynamics and longevity of circulating CD4+ memory T cells depend on the age of the cells and not the chronological age of the host"

* Please add the links to the funding agencies in the Financial Disclosure statement in the manuscript details.

DATA POLICY:

Regardless of the method selected, please ensure that you provide the individual numerical values that underlie the summary data displayed in the following figure panels as they are essential for readers to assess your analysis and to reproduce it: 2A–D, 4A, 4B, 5A–C, 6, and 7B.

CODE POLICY

Per journal policy, if you have generated any custom code during the curse of this investigation, please make it available without restrictions upon publication. Please ensure that the code is sufficiently well documented and reusable, and that your Data Statement in the Editorial Manager submission system accurately describes where your code can be found. 

We expect to receive your revised manuscript within two weeks. 

*Published Peer Review History*

*Press*

Sincerely,

Christian

Christian Schnell (on behalf of Ines who is out of office this week)

Senior Editor

cschnell@plos.org

PLOS Biology

Ines Alvarez-Garcia, PhD

Senior Editor

PLOS Biology

Reviewer remarks:

Reviewer #1: Thank you to the authors for responding to the questions and concerns in the first review.

The revised manuscript explains the results and caveats of the approaches and describes the authors' findings in the context of the literature. 

I do have two concerns on the new text/material

1. Line 254, 'measures the persistence of a TCR clonotype'. This text implies the author have tracked the CD4 T cell populations by TCR clonotype, can they re-phrase the text? The main concern here is potential confusion by readers, especially as the detailed methods used in these studies must be read in a different paper.

2. The experiments shown in the Figure 7: my understanding is that these are new experiments that have not been previously described in the 2017 Bio-protocol paper. It is important, therefore, that more complete methodology and some relevant flow plots/gating are included in this manuscript to allow readers to evaluate the method and primary data. The new data are consistent with the authors' conclusions based on the busulfan chimeras but the current limited experimental detail, especially for the transfer experiment (Fig 7A-B), mean that alternative explanations of these results are possible.

In particular, how the Tem and bulk CD4 T cells were isolated (e.g. by magnetic sorting or FACS-based sorting?) should be described. If different methods were used to sort the Tem and bulk CD4 T cells, this could be an alternative, technical explanation for the difference in proliferation described in Fig 7. Similarly, were equal number of bulk and Tem cells transferred, or the same number of Tem transferred in each population? Without these data and inclusion of some ex vivo day 7 flow plots, the reader cannot know how reliable the %Ki67-mCherry data are. Same argument for the mTomato+ cell populations. 

Additionally, it is unclear if the day 0 time point in Fig 7B shows data for the cells pre-transfer of are transferred cells examined on the same day of transfer to take account for the loss of cells following the transfer. Some additional clarity on these experiments is required. 

Reviewer #2 (Jose Borghans): I'm happy with the way the authors addressed my points and would recommend publication of the paper.

---

## [Editor Report · Decision Letter 3]

24 Jun 2024

Dear Dr Yates,

Thank you for the submission of your revised Research Article entitled "The dynamics and longevity of circulating CD4+ memory T cells depend on cell age and not the chronological age of the host" for publication in PLOS Biology. On behalf of my colleagues and the Academic Editor, Avinash Bhandoola, I am delighted to let you know that we can in principle accept your manuscript for publication, provided you address any remaining formatting and reporting issues. These will be detailed in an email you should receive within 2-3 business days from our colleagues in the journal operations team; no action is required from you until then. Please note that we will not be able to formally accept your manuscript and schedule it for publication until you have completed any requested changes.

PRESS

Sincerely, 

Ines

--

Ines Alvarez-Garcia, PhD

Senior Editor

PLOS Biology
